# Early full-day leg movement kinematics and swaddling patterns in infants in rural Guatemala: A pilot study

Jinseok Oh[1], Eva Leticia Tuiz Ordoñez[2], Elisa Velasquez[3], Marines Mejía[3], Maria del Pilar Grazioso[2,3], Peter Rohloff[2,4☯]*, Beth A. Smith[1,5,6☯]*

1 Division of Developmental-Behavioral Pediatrics, Children's Hospital Los Angeles, Los Angeles, California, United States of America, 2 Wuqu' Kawoq | Maya Health Alliance, Tecpán, Guatemala, 3 Proyecto Aigle, Guatemala City, Guatemala, 4 Division of Global Health Equity, Brigham and Women's Hospital, Boston, Massachusetts, United States of America, 5 Department of Pediatrics, Keck School of Medicine, University of Southern California, Los Angeles, California, United States of America, 6 Developmental Neuroscience and Neurogenetics Program, The Saban Research Institute, Children's Hospital Los Angeles, Los Angeles, California, United States of America

☯ These authors contributed equally to this work.
* peter@wuqukawoq.org (PR); bsmith@chla.usc.edu (BAS)

**Data Availability Statement:** Data collected for this study will be deidentified and deposited in National Institute of Child Health and Human Development

## Abstract

### Background

Tools to accurately assess infants' neurodevelopmental status very early in their lives are limited. Wearable sensors may provide a novel approach for very early assessment of infant neurodevelopmental status. This may be especially relevant in rural and low-resource global settings.

### Methods

We conducted a longitudinal observational study and used wearable sensors to repeatedly measure the kinematic leg movement characteristics of 41 infants in rural Guatemala three times across full days between birth and 6 months of age. In addition, we collected sociodemographic data, growth data, and caregiver estimates of swaddling behaviors. We used visual analysis and multivariable linear mixed models to evaluate the associations between two leg movement kinematic variables (awake movement rate, peak acceleration per movement) and infant age, swaddling behaviors, growth, and other covariates.

### Results

Multivariable mixed models of sensor data showed age-dependent increases in leg movement rates (2.16 [95% CI 0.80,3.52] movements/awake hour/day of life) and movement acceleration (5.04e-3 m/s$^2$ [95% CI 3.79e-3, 6.27e-3]/day of life). Swaddling time as well as growth status, poverty status and multiple other clinical and sociodemographic variables had no impact on either movement variable.

Data and Specimen Hub (DASH) and could be available through DASH.

**Funding:** Research reported in this publication was supported by the Eunice Kennedy Shriver National Institute of Child Health & Human Development of the National Institutes of Health (https://www.nichd.nih.gov/) under Award Number R21HD096521, awarded to BS and PR. The content is solely the responsibility of the authors and does not necessarily represent the official views of the National Institutes of Health. The funders had no role in study design, data collection and analysis, decision to publish, or preparation of the manuscript.

**Competing interests:** The authors have declared that no competing interests exist.

## Conclusions

Collecting wearable sensor data on young infants in a rural low-resource setting is feasible and can be used to monitor age-dependent changes in movement kinematics. Future work will evaluate associations between these kinematic variables from sensors and formal developmental measures, such as the Bayley Scales of Infant and Toddler Development.

## Introduction

A pressing global need is supporting optimal growth and development for all infants. More than 40% of children under age 5 residing in low- and middle-income countries (LMICs) are reported to be at risk of not reaching their developmental potential [1,2]. Multiple factors including malnutrition, poverty, and lack of early stimulation a contributory [3]. Early developmental monitoring and support, particularly during the first few years of life are effective [4]; however, in resource-limited settings early and accurate identification of which infants are most at need for early intervention is a critical problem.

Among LMICs, Guatemala has one of the proportionally largest populations of infants at risk for impaired growth and development. This is especially the case for Indigenous Maya infants. The Maya population constitute more than 40% of the country's population and resides disproportionately in rural areas with less access to health, sanitation, and early education infrastructure. In addition, Maya children demonstrate one of the highest rates of chronic malnutrition or stunting (low height-for-age) in the world [5]. Although many interventions have been developed to support growth and development for populations with high rates of stunting, few focus on identifying developmental risk and providing development support in the first few months of infancy, before the adverse cumulative impacts of malnutrition and other environmental exposures have accumulated and when the opportunities to foster optimal development are the greatest [6,7].

A limiting factor in LMICs for very early intervention targeting to infants at risk of impaired growth and development is a lack of sensitive and feasible tools that can detect impaired growth and development at a very early age. Current clinical assessments used in LMICs include monitoring length for the onset of stunting, which is technically difficult with only subtle loss of height velocity typical prior to the onset of complementary feeding at 6 months of age. Individual developmental monitoring and home environment assessments require special training and extensive cultural adaptation and can be time consuming; are typically more heterogenous and less precise in early infancy; and are only moderately predictive of later neurodevelopmental outcomes [8–10]. Improved tools that can better identify which infants are developing typically within the first few months of life–a critical period in infant development–are badly needed.

Collecting and analyzing comprehensive infant movement data using wearable sensors to predict developmental status is one potential novel solution to this problem. The theoretical ground for this method is that movement control is a dominant component of neonatal behavior assessments in both healthy infants and infants at-risk for developmental difficulties. Observation of certain movement characteristics, or lack thereof, could assist in characterizing and predicting emerging subclinical developmental difficulties. As one example, infants with cerebral palsy display an early lack of fidgety movements at 3–4 months post-term age [11,12].

Further, wearable sensors can provide more objective, quantitative measures, such as the duration or intensity of movement, measured across longer periods of time than are typically

observed during traditional short assessments. In addition to being based on subjective observation, traditional short clinical assessments only reflect the movement behavior of the infant at a moment of time, as opposed to capturing their full repertoire of movement behaviors across a full day as is possible with wearable sensors. A study using accelerometer recordings over 48 hours found that infants with Down syndrome spent more time generating leg activities of low intensity, compared to infants with typical development [13]. Previously, we compared values of kinematic variables extracted from full-day recordings of wearable sensors between infants with typical development and infants at risk for developmental disabilities. Infants with poor developmental outcomes at 24 months of age exhibited significantly lower values of kinematic variables in early infancy (e.g., peak acceleration per movement) compared both to infants at risk but with good developmental outcomes and infants with typical development [14]. Longitudinal data from wearable sensors further enable measures of the complexity or the variability of the signal magnitude, such as entropy. Smith et al [15] calculated sample entropy from wearable sensor data and showed that infants at risk of developmental difficulty had significantly lower values of sample entropy than infants with typical development. Taken together, these empirical findings support the potential for early use of wearable sensors to anticipate later developmental outcomes of infants.

The main goal of the current study was to measure the kinematic movement traits of infants in rural Guatemala at risk for developmental difficulties due to poverty, malnutrition, rural location, and other environmental factors. In this manuscript, we report kinematic characteristics of infant leg movements collected longitudinally across 3 months between birth and 6 months of age, and explored whether covariates including swaddling, growth, and sociodemographic characteristics impacted leg movement rate or acceleration (intensity) of leg movements. Swaddling is the common practice in rural Guatemala and many other settings of keeping the infant tightly wrapped in fabric while sleeping and also, frequently, when carried on the caregiver's back when performing chores or when out of the house. This is formative work that will allow future planned analyses correlating early infancy kinematic characteristics with later neurodevelopmental assessments.

## Material and methods

### Participants and recruitment procedures

Institutional Review Board approval was obtained from Maya Health Alliance (WK 2019 004), the University of Southern California (HS-19-00564), and Children's Hospital Los Angeles (CHLA-20-00201). Subjects were recruited in Tecpán, Guatemala, a rural majority Indigenous Maya town, after referral by providers in routine clinical settings operated by Maya Health Alliance, a primary healthcare organization with a major clinical center in the study community. Verbal informed consent from one parent or legal guardian was obtained by a study staff witnessed by a second study staff. The script used to provide verbal consent was signed by the administering study staff member and filed, and a second copy given to the study participant. Verbal consent is frequently used by Maya Health Alliance for minimal risk studies as signatures are not culturally common in Indigenous Maya communities. This procedure was approved by the IRB's. Study staff were fluent in both Spanish and Kaqchikel Maya, the two primary languages spoken in the region. Inclusion criteria were: parent or legal guardian willing and able to provide consent; infant aged 0–16 weeks at enrollment, singleton birth, full-term (>38 weeks) birth. Exclusion criteria were: presence of acute malnutrition/wasting (weight for length more than 2 standard deviations below the WHO Child Growth Reference Population) or severe medical illness including heart disease, kidney disease, congenital abnormality, genomic syndromes and severe neurological deficits as determined by Maya Health

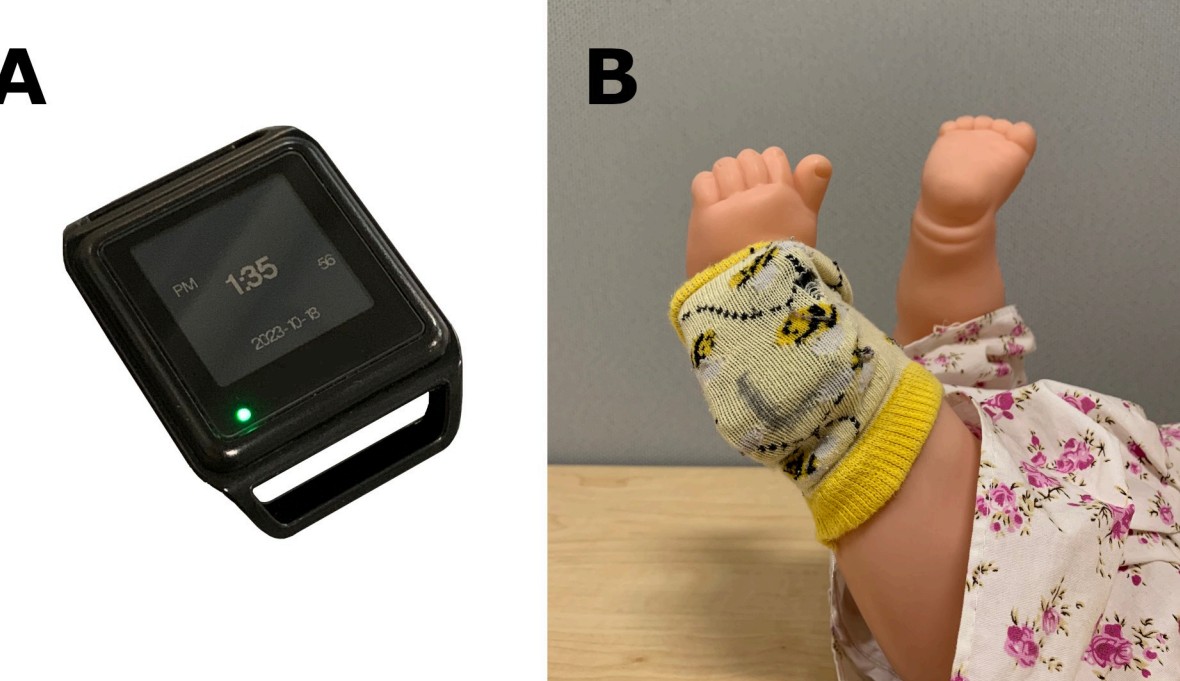

**Fig 1. Opal sensors worn on an infant.** (A) A wearable sensor used in each recording. (B) Custom-made legwarmers with sensors inserted put on each ankle of an infant.

Alliance clinicians. This was an exploratory formative scientific study of leg movement characteristics in this population, and therefore no sample size calculation was performed. We decided a priori that a sample of 40 infants would be sufficient for testing for associations and determining effect sizes for future studies based on a recently completed clinical trial assessing infants across 6 months at Maya Health Alliance [16]. Participants were recruited during the period: 2/18/2020–10/14/2021.

## Apparatus

Opal (version 2) sensors (APDM Inc., Portland, OR, USA), inserted into pockets of custom-made legwarmers, were worn on each ankle of an infant (Fig 1). The size of the pocket was designed specifically to hold the sensors in place, and a variety of sizes of legwarmers to choose from provided a close but not too tight fit for each infant. The dimension of a sensor was 43.7 x 39.7 x 13.7 mm, and the weight was less than 25 g. The sensors were actively synchronized in time, recording at the sampling rate of 20 Hz. The validity of measuring the quantity, duration, and acceleration of infant leg movements has been reported previously [17,18].

## Procedure

Study assessments occurred in the morning either within subjects' homes or the Maya Health Alliance clinic. Sensors were placed on both legs by study staff at each visit, at which point caregivers were instructed to resume their typical daily activities. Movement sensors were worn during three separate days from recruitment to 6 months of age at 1-month (± 1 week) intervals. Each time, the infant wore the sensors from the morning placement until bedtime (10–12 hours). Sensors could be removed briefly while engaged in any water activities (e.g.,

bathing or swimming). Caregivers then removed the sensors from the infant at bedtime, and they were collected afterward by research staff.

At each visit, growth data (infant length and weight) were collected, and caregivers answered survey questions about how long the infant was swaddled while lying down or worn swaddled against the caregiver typically in a day and specifically during the time the sensors were worn. Survey and growth data was collected by trained bilingual (Spanish/Kaqchikel) research technicians. At the first visit, additional sociodemographic and medical history data were also collected.

## Measurements

**Quantitative measures of leg movement kinematics.** Full-day leg movement data were analyzed using the method introduced previously [17,18]. First, the daily leg movement rates (movements per hour awake) were calculated from each leg (left and right) and the average calculated. Peak acceleration per movement ($m/s^2$) was calculated as well from each leg and summarized for both legs as the median peak acceleration across all movements. Sleep time was defined as the sum of all 5-minute periods that showed less than 3 movements. When sensors were removed, caregivers provided estimates of four measures regarding their swaddling behavior: 1) number of hours the infant was wrapped and laid down while the sensors were on, 2) number of hours the infant was wrapped against the caregiver while the sensors were on, 3) number of hours the infant is typically wrapped and laid down in a usual day, and 4) number of hours the infant is typically wrapped against the caregiver in a usual day. Total swaddled time during the study was defined as the sum of the first two variables. Total swaddle time during a typical day was defined as the sum of the last two variables.

## Statistical analysis

**Linear mixed model analysis to explore variability of leg movement rate.** To explore the between-subject variability of collected daily leg movement rate data, 7 non-kinematic variables were selected from review of literature and consensus among the authors of factors likely associated with infant motor development. These included age in days at each study visit, birth weight, sex assigned at birth, caregiver report of any pregnancy or perinatal complication, length-for-age z-score calculated using the World Health Organization Child Growth Reference Standards (LAZ), raw poverty score derived from a validated local asset survey (Poverty Probability Index), and caregiver-reported hours the infant was swaddled while sensors were on. A linear mixed model with these seven fixed effects and the random effect of individual infant was fitted to the daily leg movement rate data. The structure of the covariance matrix was compound symmetry. Subsequently, nonsignificant fixed effects were removed from the initial model based on serial likelihood ratio tests for nested models. The normality of the residual distribution as well as the random effect distribution was tested with Shapiro-Wilk's test. All analyses were done with STATA and lme4 package of R (version 4.2.2).

## Results

### Summary of recruitment and participant characteristics

Forty-nine infants were recruited from a single health district (Tecpán, Chimaltenango) in Guatemala by research staff at Maya Health Alliance. Data collection occurred between February 2020 and January 2022; however, the study was paused for 10 months due to the COVID-19 pandemic. A summary of subject flow through the study is given in Fig 2. As shown in

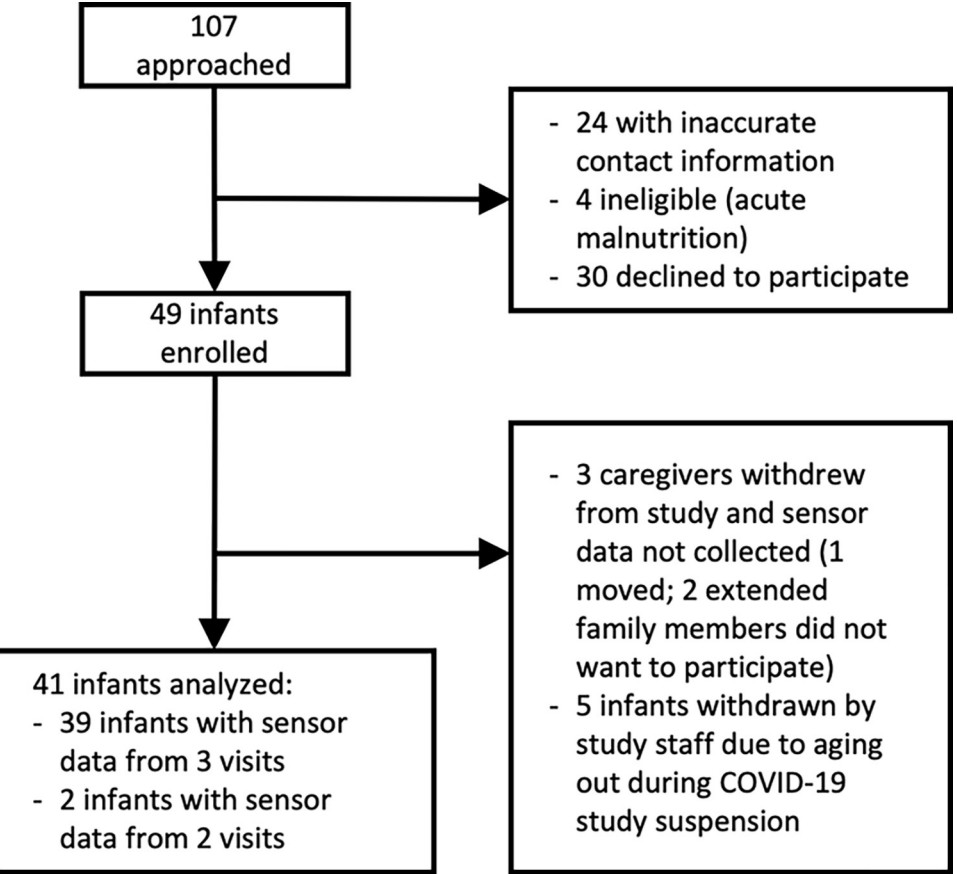

**Fig 2. Flow diagram of subject participation in a study of wearable sensors to assess leg movement in infants in rural Guatemala.**

Fig 2, data from a total of 41 infants were included in the analysis presented here. Table 1 describes the baseline characteristics of participants.

## Growth and leg movement data of infants

Table 2 shows the z-scores for growth and movement characteristics of the 41 infants who formed the final sample. Median length-for-age z-scores (LAZ) were lower than median scores for weight-for-age (WAZ) and weight-for-length (WLZ) at all three visits. LAZ scores were below -1 Z-scores at the first visit and decreased over the course of the study, consistent with the known phenomenon of infants in rural Guatemala born short and the emergence of linear growth faltering over the first year of life in a setting of endemic stunting [19]. Regarding the movement data, all kinematic variables (leg movement rate, median peak acceleration per movement of each leg) showed increases with age across the visits (see Fig 3).

## Description of infant swaddling characteristics

Table 3 describes the caregiver-estimated hours infants were swaddled either when sensors were on or during a typical day. All six variables had right-skewed distributions. To check that infants were swaddled similarly when sensors were on compared to a typical day, we conducted Wilcoxon signed rank tests. Total estimated swaddling hours, in all three visits, did not significantly differ between days when sensors were in place and the typical day condition.

**Table 1. Baseline characteristics of participants (N = 41).**

| Characteristic | % or Median (IQR) |
|---|---|
| **Sex assigned at birth (Female)** | 51% |
| **Ethnicity (Indigenous Maya)** | 95% |
| **Age in days at first sensor visit** | 63 days (44–88) |
| **Primary caregiver** | |
| Biological mother | 98% |
| Primary school education or less | 68% |
| **Prenatal and perinatal history** | |
| Received prenatal care | 100% |
| Any complication during pregnancy | 30% |
| Any complications during childbirth | 12% |
| Delivered in the home | 39% |
| Caesarean section | 54% |
| Birth weight | 3.1 kg (2.8–3.5) |
| **Raw poverty score [a]** | 35 (26–45) |

[a] Raw poverty score can range 0–100. A raw score of 35–39 corresponds to a 66% probability of living below the 100% national poverty line; lower scores indicate more poverty.

**Table 2. Growth and movement characteristics of infants per visit.**

| Visit | age (days) | | LAZ | | WAZ | | WLZ | | Movement rate (mov/hour) | | Left peak acc/mov (m/s²) | | Right peak acc/mov (m/s²) | |
|---|---|---|---|---|---|---|---|---|---|---|---|---|---|---|
| | M | IQR | M | IQR | M | IQR | M | IQR | M | IQR | M | IQR | M | IQR |
| 1 | 63 | 44 | -1.27 | 1.44 | -0.5 | 1.71 | 0.78 | 1.15 | 930 | 464 | 2.78 | 0.47 | 2.65 | 0.38 |
| 2 | 94 | 40 | -1.53 | 1.42 | -0.54 | 1.73 | 0.73 | 1.18 | 1005 | 275 | 2.91 | 0.42 | 2.88 | 0.39 |
| 3 | 129 | 35.5 | -1.55 | 1.76 | -0.58 | 1.55 | 0.58 | 1.07 | 1040 | 431 | 3.06 | 0.31 | 3.00 | 0.38 |

LAZ, length-for-age Z score; WAZ, weight-for-age Z score; WLZ, weight-for-length Z score; M, median; IQR, interquartile range.

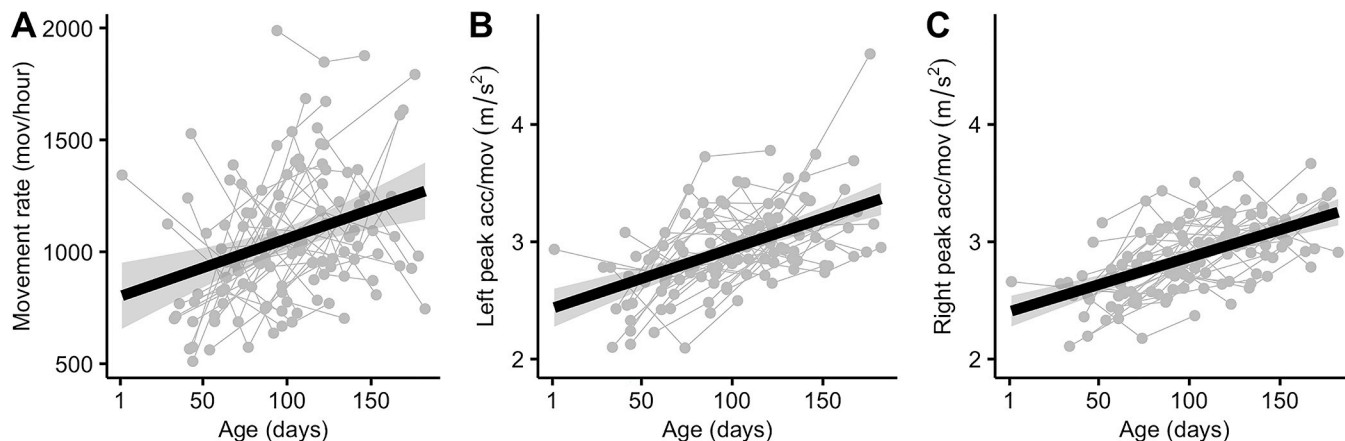

**Fig 3. Age dependent change of the movement characteristics.** (A) Leg movement rate, (B) left leg peak acceleration per movement, and (C) right leg peak acceleration per movement are plotted against age at each recording. Age is in days since birth. Connected gray dots are estimated hours of infant swaddling reported for each infant across three visits. Thick black lines are linear estimation of the age dependent trends. Shading around the lines indicate 95% confidence intervals.

**Table 3. Estimated hours of infant swaddling at each study visit.**

| Visit | While sensors were on | | | | | | During a typical day | | | | | |
|---|---|---|---|---|---|---|---|---|---|---|---|---|
| | Swaddled and laid down | | Swaddled against caregivers | | Total | | Swaddled and laid down | | Swaddled against caregivers | | Total | |
| | M | IQR | M | IQR | M | IQR | M | IQR | M | IQR | M | IQR |
| 1 | 0.5 | 3 | 2 | 2.5 | 2 | 5.5 | 0.5 | 3 | 1 | 2.5 | 3 | 5.25 |
| 2 | 0 | 1 | 1 | 1.5 | 2 | 3 | 0 | 1.5 | 0.75 | 2 | 2 | 3.5 |
| 3 | 0 | 0 | 2 | 2.7 | 2 | 2.3 | 0 | 0 | 2 | 2.7 | 2 | 3 |

M, median; IQR, interquartile range.

Fig 4 displays age-dependent changes in daily estimated hours infants were swaddled. Locally estimated scatterplot smoothing (LOESS) curves fitted to total time swaddled (thick black curves in Fig 4A or 4D) show age-associated decreases, particularly between 0 and 100 days from birth. These decreases in infant swaddling were largely driven by decreases in swaddling time while laid down, as the amount of time spent swaddled against the caregiver was consistent over the entire study period (median age 129 days at last study visit; compare Fig 4A–4F).

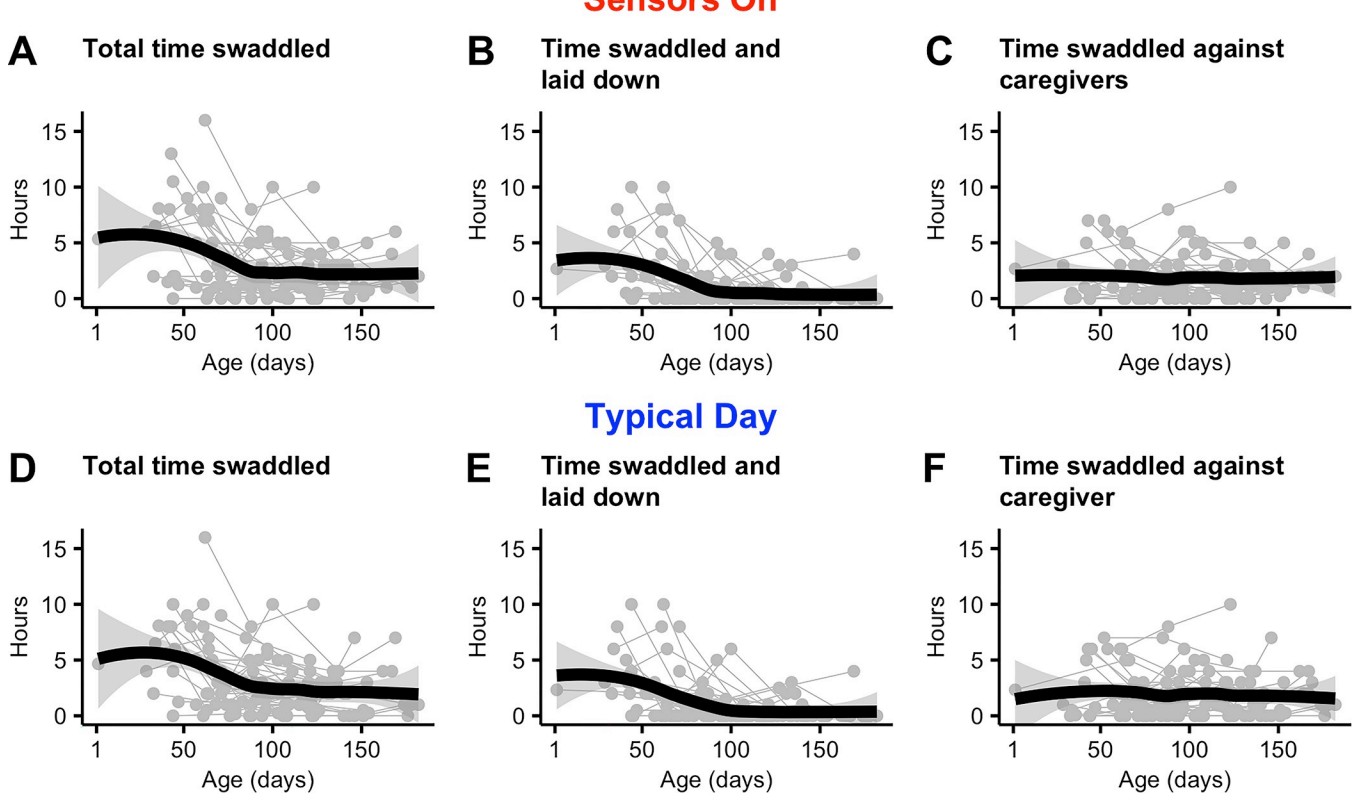

**Fig 4. Age dependent change in hours of infant swaddling.** The top row shows caregiver estimated hours infants were swaddled (A) in total, (B) laid down on the floor, or (C) against caregivers during a visit (sensors on). The bottom row displays analogous measure (D) in total, (B) when swaddled and laid down, and (C) when swaddled against caregivers. Total swaddle time is the sum of time swaddled and laid down and time swaddled against caregivers. Thick black curves are LOESS (locally estimated scatterplot smoothing) curves fitted to data to estimate age-associated change of swaddling hours. Shading around the curves indicate 95% confidence intervals.

**Table 4. Summary of models on leg movement rates.**

| Predictors | Model 1: Movement rate, fully specified | | | Model 2: Movement rate, parsimonious model | | |
|---|---|---|---|---|---|---|
| | Estimates | 95% C.I. | p | Estimates | 95% C.I. | p |
| (Intercept) | 1570.13 | 848.55–2291.70 | **<0.001** | 900.21 | 727.88–1072.54 | **<0.001** |
| Age (days) | 1.94 | 0.37–3.51 | **0.016** | 2.16 | 0.80–3.52 | **0.002** |
| Sex (Female = 0; Male = 1) | -160.20 | -317.52–-2.87 | **0.046** | -115.33 | -245.43–14.77 | 0.082 |
| Birth weight (kg) | -169.98 | -378.95–38.99 | 0.110 | | | |
| Any obstetric/perinatal complication | -21.98 | -168.97–125.01 | 0.767 | | | |
| Length-for-age Z score (LAZ) | 58.47 | -5.66–122.60 | 0.073 | | | |
| Raw poverty score | 0.24 | -5.59–6.08 | 0.934 | | | |
| Total swaddle time (hour) | -4.35 | -24.37–15.67 | 0.667 | | | |
| **Random Effects** | | | | | | |
| $\sigma^2$ | 53826.11 | | | 59879.99 | | |
| $\tau_{00\ infant}$ | 26539.00 | | | 22222.13 | | |
| ICC | 0.33 | | | 0.27 | | |
| N $_{infant}$ | 35 | | | 41 | | |
| Observations | 102 | | | 120 | | |
| Marginal $R^2$ / Conditional $R^2$ | 0.178 / 0.449 | | | 0.129 / 0.365 | | |

C.I. = Confidence Interval; σ2 = residual variance; τ00 infant = variance of random effect; ICC = Intraclass Correlation Coefficient; N infant = Number of infants; Fully specified model has fewer observations due to missing values.

### Full-day leg movement rate is dependent only on age

Linear multivariable mixed models were used to explore variables impacting full-day leg movement rates. Age in days significantly impacted leg movement rates. The age-associated increase of the movement rate is demonstrated in Fig 3A. As infants grew, movement rates increased (2.16 [95% CI 0.80,3.52] movements/hour awake per day of life, Table 4, model 2). A fully specified model including other covariates (Table 4, model 1) did not perform significantly better than the model containing just age and sex assigned at birth.

### Leg movement intensity is dependent on age

In addition to the leg movement rate, infants' peak acceleration per movement also increased with age (see Fig 3B and 3C). Linear mixed models were again used to examine this trend with the average of the measures from each leg as the dependent variable. A fully specified model with covariates (Table 5, model 3) again did not perform significantly better than the simplified model adjusted just for age and sex assigned at birth (Table 5, model 4). The peak acceleration increased with 5.04e-3 m/s$^2$ [95% CI 3.79e-3, 6.27e-3] per day of age.

### Correlation between sleep time estimated from sensor data and caregiver-reported swaddling time

Sleep time was estimated from sensor data (defined as the sum of all 5-minute periods that showed less than 3 limb movements). Fig 5 plots correlations between sensor-estimated sleep time and caregiver estimated swaddling time at each visit. Visual inspection shows gradual decreases in sleep time with age, associated with corresponding decreases in amount of time infants are swaddled while laid down (Fig 5A–5C). On the other hand, time spent swaddled against the caregiver did not decrease, even while sleep time decreased (Fig 5D–5F).

**Table 5. Summary of models on peak acceleration per movement.**

| Predictors | Model 3: Average peak acceleration, fully specified | | | Model 4: Average peak acceleration, parsimonious model | | |
|---|---|---|---|---|---|---|
| | *Estimates* | *95% C.I.* | *p* | *Estimates* | *95% C.I.* | *p* |
| (Intercept) | 2.72 | 2.13–3.32 | **<0.001** | 2.39 | 2.24–2.55 | **<0.001** |
| Age (days) | 4.74e-3 | 0.00–0.01 | **<0.001** | 5.04e-3 | 0.00–0.01 | **<0.001** |
| Sex (Female = 0; Male = 1) | -0.07 | -0.19–0.06 | 0.284 | 0.02 | -0.09–0.14 | 0.703 |
| Birth weight (kg) | -0.03 | -0.20–0.13 | 0.684 | | | |
| Any obstetric/perinatal complication | -0.06 | -0.18–0.06 | 0.304 | | | |
| Length-for-age Z score (LAZ) | 0.01 | -0.04–0.07 | 0.642 | | | |
| Raw poverty score | -3.16e-3 | -0.01–0.00 | 0.174 | | | |
| Total swaddle time (hour) | -3.58e-3 | -0.02–0.01 | 0.696 | | | |
| **Random Effects** | | | | | | |
| $\sigma^2$ | 0.05 | | | 0.05 | | |
| $\tau_{00\ infant}$ | 0.01 | | | 0.02 | | |
| ICC | 0.18 | | | 0.24 | | |
| $N_{infant}$ | 35 | | | 41 | | |
| Observations | 102 | | | 120 | | |
| Marginal $R^2$ / Conditional $R^2$ | 0.402 / 0.507 | | | 0.347 / 0.502 | | |

Fully specified model has fewer observations due to missing values.

# Time swaddled and laid down

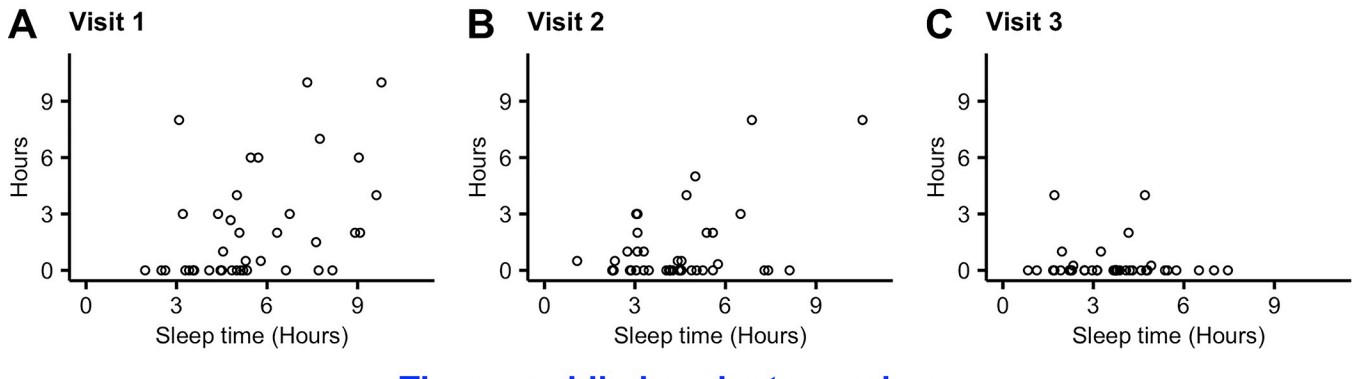

# Time swaddled against caregivers

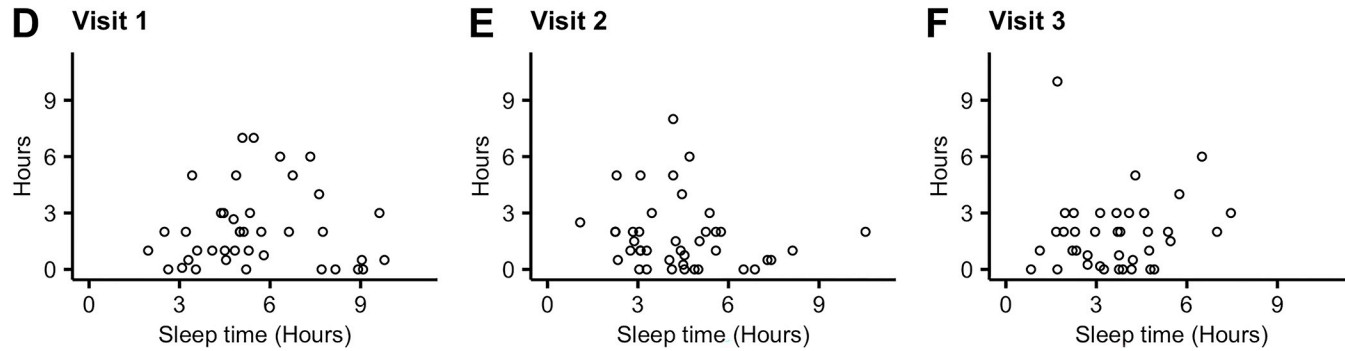

**Fig 5. Two types of time infants were swaddled against sleep time estimated by the algorithm at each visit.** Plots A-C show time swaddled and laid down against sleep time (periods of inactivity) at visit 1, 2, and 3. Plots D-F show time swaddled against caregivers against sleep time at visit 1, 2, and 3 respectively.

## Discussion

A major goal of this research is to develop new, feasible tools for very early monitoring of infant development in low-resource settings. Using visual analysis and linear mixed models we show that sensor kinematic data can be used to derive limb movement and acceleration rates in this population. The first main finding of the present study is that multiple adjustments for growth status, poverty, and other clinical and sociodemographic covariables were not significant, with limb movement rates being largely determined by age alone. This result in itself is interesting, because it suggests that early infant motor development may not be strongly influenced by many of the factors commonly thought to influence later neurodevelopmental outcomes, or that movement rate alone is not sufficient to describe motor development, and what infants are doing when they are moving is as important to consider as how much they are moving [17]. In forthcoming analyses, we will further explore this finding by examining associations between limb movement characteristics and formal neurodevelopmental assessments (Bayley Scales of Infant and Toddler Development) at later ages.

The second main finding of this study is that caregiver-reported infant swaddling is not correlated with age-dependent increases in limb movement. An important component of this formative research was collecting information on infant swaddling behaviors. As is common in many cultures worldwide, it is well known that Indigenous Maya infants in rural Guatemala are frequently swaddling, both when laid down as well as carried against the caregiver's body when performing domestic work or leaving the home. However, quantification of this care behavior has been lacking. Here we report age-dependent decreases in infant swaddling time, driven mostly by swaddling while laid down. Age-dependent decreases in time swaddled while lying down were correlated with age-dependent decreases in sleep time, reinforcing the idea that swaddling is a care behavior in part used to contain young infants while they sleep. On the other hand, time spent swaddled against the caregiver did not decrease with age, presumably because infants continue to be swaddled in this fashion for some portion of awake time and when the caregiver engages in domestic activities or leaves the home.

Importantly, there was no association between swaddling time and limb movement kinematic variables. Our study therefore provides novel motor-kinematic data corroborating other observational data on the neutral to positive developmental impacts of infant swaddling in the global literature. For example, preterm infants in high-income acute care settings showed better motor organization when swaddled than when not [20]. Another study on a similar group of infants showed that swaddling improved muscle tone and sleep quality [21]. Mongolian infants who were swaddled from birth to 7 months scored similarly on the Bayley Scales of Infant Development, Second Edition (BSID-II) compared to their un-swaddled peers [22].

## Strengths and limitations

The strengths of this pilot study include 1) the study recruited a sample of largely Indigenous Maya infants from rural Guatemala and 2) we used a validated wearable sensor algorithm to measure infant leg movements across days in the natural environment.

The limitations of this pilot study include that we recruited a small convenience sample of a rural Indigenous Maya population in Guatemala and therefore generalizability may be limited both to other communities in Guatemala and globally. We describe our method as feasible given our high rate of success with wearable sensor data collection and analyses, however we did not a priori define a threshold for determining that the method was feasible. In addition, one of the key variables analyzed here, swaddling time, was estimated by caregiver recall; this recall method may be of limited precision and corroboration by direct observation may be indicated in future studies. Individual level variability among infants in key covariables, such

as nutritional status, may have obscured the magnitude of some findings in this small sample size; we mitigated this by using linear mixed models includes random effects for individuals and fixed effects for key covariables. Finally, sensor data reflects primary limb motor development, which may not be directly correlated either with overall infant motor development or with infant development in other domains (language, cognition). In forthcoming analysis, we will address this limitation by examining correlations between kinematic data and formal neurodevelopmental assessments (Bayley Scales of Infant and Toddler Development).

## Conclusions

Earlier higher-fidelity monitoring techniques to identify infants at risk of developmental difficulties are needed, especially in lower-resource settings. Wearable sensors are one promising tool that may allow for non-invasive monitoring of infant motor development with minimal need for training and expertise. In this study, we show that sensors could be used to monitor age-dependent changes in leg movement kinematic data from a population of rural infants in Guatemala. These age-dependent changes were not significantly impacted by swaddling time or other major clinical or sociodemographic variables. In future work we will investigate how kinematic measures collected with wearable sensors relate to these infants' later neurodevelopmental outcomes, including not just motor but also cognitive and language development, and how these interact with social determinants of health including poverty, home care environments, and access to opportunities for early education.

## Author Contributions

**Conceptualization:** Maria del Pilar Grazioso, Peter Rohloff, Beth A. Smith.

**Data curation:** Eva Leticia Tuiz Ordoñez, Elisa Velasquez, Marines Mejía, Maria del Pilar Grazioso.

**Formal analysis:** Jinseok Oh, Peter Rohloff.

**Funding acquisition:** Peter Rohloff, Beth A. Smith.

**Investigation:** Peter Rohloff, Beth A. Smith.

**Methodology:** Maria del Pilar Grazioso, Beth A. Smith.

**Project administration:** Eva Leticia Tuiz Ordoñez, Elisa Velasquez, Marines Mejía, Maria del Pilar Grazioso.

**Resources:** Eva Leticia Tuiz Ordoñez, Elisa Velasquez, Marines Mejía.

**Supervision:** Peter Rohloff, Beth A. Smith.

**Visualization:** Jinseok Oh.

**Writing – original draft:** Jinseok Oh, Peter Rohloff, Beth A. Smith.

**Writing – review & editing:** Jinseok Oh, Eva Leticia Tuiz Ordoñez, Elisa Velasquez, Marines Mejía, Maria del Pilar Grazioso, Peter Rohloff, Beth A. Smith.

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
