## [Decision Letter · Decision Letter 0]

16 Oct 2023

PONE-D-23-20339Early full-day leg movement kinematics and swaddling patterns in infants in rural GuatemalaPLOS ONE

Dear Dr. Smith,

Thank you for submitting your manuscript to PLOS ONE. After careful consideration, we feel that it has merit but does not fully meet PLOS ONE’s publication criteria as it currently stands. Therefore, we invite you to submit a revised version of the manuscript that addresses the points raised during the review process.

We look forward to receiving your revised manuscript.

Kind regards,

Tadashi Ito

Academic Editor

PLOS ONE

2. In the ethics statement in the Methods, you have specified that verbal consent was obtained. Please provide additional details regarding how this consent was documented and witnessed, and state whether this was approved by the IRB

3. Please include a complete copy of PLOS’ questionnaire on inclusivity in global research in your revised manuscript. Our policy for research in this area aims to improve transparency in the reporting of research performed outside of researchers’ own country or community. The policy applies to researchers who have travelled to a different country to conduct research, research with Indigenous populations or their lands, and research on cultural artefacts. The questionnaire can also be requested at the journal’s discretion for any other submissions, even if these conditions are not met.  Please find more information on the policy and a link to download a blank copy of the questionnaire here: https://journals.plos.org/plosone/s/best-practices-in-research-reporting. Please upload a completed version of your questionnaire as Supporting Information when you resubmit your manuscript.

6. We note that Figure 1 in your submission contain copyrighted images. All PLOS content is published under the Creative Commons Attribution License (CC BY 4.0), which means that the manuscript, images, and Supporting Information files will be freely available online, and any third party is permitted to access, download, copy, distribute, and use these materials in any way, even commercially, with proper attribution. For more information, see our copyright guidelines: http://journals.plos.org/plosone/s/licenses-and-copyright.

Reviewers' comments:

Reviewer's Responses to Questions

**Comments to the Author**

1. Is the manuscript technically sound, and do the data support the conclusions?

Reviewer #1: Yes

Reviewer #2: Yes

Reviewer #3: Yes

2. Has the statistical analysis been performed appropriately and rigorously? 

Reviewer #1: Yes

Reviewer #2: I Don't Know

Reviewer #3: I Don't Know

3. Have the authors made all data underlying the findings in their manuscript fully available?

Reviewer #1: Yes

Reviewer #2: Yes

Reviewer #3: Yes

4. Is the manuscript presented in an intelligible fashion and written in standard English?

Reviewer #1: Yes

Reviewer #2: Yes

Reviewer #3: No

5. Review Comments to the Author

Reviewer #1: an interesting topic and the manuscript get some significant finding. However, we have several concerns about this manuscript.

Title needs to be modified. “Pilot Study” should appear in the title.

Abstract:

Trial design is not mentioned.

Methods:

How was sample size determined?

Discussion:

1. The discussion section needs to be described scientifically. Kindly frame it along the following lines:

i. Main findings of the present study

ii. Comparison with other studies

iii. Implication and explanation of findings

iv. Strengths

Reviewer #2: From my point of view, this paper addresses a type of new technology in assessing leg movement kinematics in infants, which may be so important especially for infants at risk of impaired growth and development. This paper demonstrated that age is associated with leg movement rates and movement acceleration. This paper may open the door for new research about using wearable sensors to detect growth and development impairments.

I would like to thank the authors for their work.

Reviewer #3: The reviewer thank the authors for their effort.

Please add a paragraph in the introduction section to indicate the term "swaddling"

you should write about the validity and reliability of the used sensors.

Please, write more details about precautions while wearing the sensors and how they are stabilized.

Why you depend upon evaluating the movement of lower limb and neglected the movement of upper limbs. If you didn't have an explanation , please write that in limitation of the study.

Also, The nature of nutrition and its variance between infants should be mentioned in the limitation of the study.

You mentioned that "infants with medical illness were excluded from the study, please give examples.

6. PLOS authors have the option to publish the peer review history of their article (what does this mean?). If published, this will include your full peer review and any attached files.

Reviewer #1: No

Reviewer #2: No

Reviewer #3: **Yes: **Asmaa* M. Elbandrawy

---

## [Author Response · Author response to Decision Letter 0]

21 Dec 2023

[This is also attached as 'Rebuttal_Letter.docx']

The authors would like to thank the reviewers for their fair reviews and detailed list of constructive suggestions. We greatly appreciate the opportunity to submit a minor revision of our paper for your renewed consideration. Below we specify how we met journal requirements and respond to reviewers’ comments and point to the corresponding changes we made in the paper.

Journal requirements

1. Please ensure that your manuscript meets PLOS ONE’s style requirements, including those for file naming.

>> We have reviewed the style requirements and updated the manuscript. We have also regenerated figures and saved them as ‘Fig1.tiff’, ‘Fig2.tiff’ and etc.

2. In the ethics statement in the Methods, you have specified that verbal consent was obtained. Please provide additional details regarding how this consent was documented and witnessed, and state whether this was approved by the IRB.

>> Verbal consent was approved by the IRB. Further details are now provided in the methods of the paper. 

3. Please include a complete copy of PLOS’ questionnaire on inclusivity in global research in your revised manuscript. 

>> Please see attached questionnaire.

4. Note on depositing data. 

>> De-identified data will be deposited in the NIH DASH repository once all analyses are completed.

5. Request for the relevant accession numbers or DOIs necessary to access your data. 

>> As noted above, deidentified data will be deposited in a public repository once all analyses are completed, but these are still ongoing. Therefore, we have changed these details for the time being to: “data is available from the corresponding authors upon reasonable request.”

6. Figure 1 contains copyrighted images.

>> A new Fig 1 is prepared. As for the image of the sensor used in the study, we took picture of the sensor we purchased. We inquired the manufacturer (APDM WEARABLE TECHNOLOGIES INC.) about using an image of the sensor and received a response from its legal department that any publicly available image does not need its permission including images on the website. I attach the email communication that confirms the use of the image (APDM_Opal_sensor_image_use_permission.pdf)

7. Please review your reference list to ensure that it is complete and correct.

>> Reference 1 has erratum but was not retracted. The reference is most up to date with the correction made (on 9/17/2018). The correction below was already reflected in the reference we included:

The following sentence should have been included in the Acknowledgments section of this Article: “We acknowledge the contribution of Charles R J Newton (University of Oxford, Oxford, UK) to an earlier version of this article.”

Reviewer 1’s concerns:

1. Title needs to be modified. “Pilot Study” should appear in the title.

>> The revised title now indicates that this is a pilot study.

2. Abstract: trial design is not mentioned

>> The revised abstract (Methods) now clarifies that we conducted a longitudinal observational study.

3. Method: How was sample size determined?

>> As stated in the manuscript, “we decided a priori that a sample of 40 infants would be sufficient for testing for associations and determining effect sizes for future studies based on a recently completed clinical trial assessing infants across 6 months at Maya Health Alliance [16] .” In previous work, we found large effect sizes (Cohen’s d = 0.8 – 1.3) for differences in leg movement quantity and acceleration between infants with typical development and infants broadly at risk for developmental delay. Here, we used G*Power to determine that with a sample size of 30 infants we would be able to detect a difference across 6 months between children of one-third to one-half of SD with 80% power, alpha = 0.05 and intra-individual correlations of 0.5-0.8.

4. Discussion: The discussion section needs to be described scientifically. Kindly frame it along the following lines:

>> a. The Discussion section now does not have sub-headings. Originally the first two subheadings were the main findings of the study. Instead, it is now more explicitly stated as below:

“The first main finding of the present study is that …” (3rd sentence of the first paragraph)

“The second main finding of this study is that …” (1st sentence of the second paragraph)

>> b. In the first paragraph, a new citation [17] was inserted to further illustrate one interpretation of the result. In the second paragraph, comparisons were already made (citations 20, 21 and 22).

>> c. Implication and explanation of finding were also provided in the original manuscript at the following positions:

- 4th sentence of the first paragraph

- last half of the second paragraph

>> d. Strengths and limitations are now summarized after the Discussion section.

Reviewer 2’s concerns: NA

Reviewer 3’s concerns:

1. Please add a paragraph in the introduction section to indicate the term “swaddling”.

>> The last paragraph of the Introduction lists swaddling as one covariate. We added an additional sentence in the last paragraph better defining swaddling: Swaddling is the common practice in rural Guatemala and many other settings of keeping the infant tightly wrapped in fabric while sleeping and also, frequently, when carried on the caregiver’s back when performing chores or when out of the house. 

2. You should write about the validity and reliability of the used sensors.

>> We have added this sentence (with the relevant citations) at the end of the Apparatus paragraph in Methods: “The validity of measuring the quantity, duration, and acceleration of infant leg movements has been reported previously.”

3. Please, write more details about precautions while wearing the sensors and how they are stabilized.

>> We have inserted this sentence to the Apparatus paragraph in Methods: “The size of the pocket was designed specifically to hold the sensors in place, and a variety of sizes of legwarmers to choose from provided a close but not too tight fit for each infant.”

4. Why you depend upon evaluating the movement of lower limb and neglected the movement of upper limbs. If you didn’t have an explanation, please write that in limitation of the study.

>> There is no reason to expect significant differences in overall movement quantity or characteristics of lower limbs compared to upper limbs in these infants. As wearing 4 sensors would be unnecessarily burdensome for the participants and require many more resources, we opted to focus on quantifying the movement of lower limbs as a representation of neuromotor control. Lower limb movement in early life has been reported to be associated with later walking onset of infants, making it one reasonable measure for the neurodevelopmental status of the infants. 

5. Also, the nature of nutrition and its variance between infants should be mentioned in the limitation of the study.

>> We have added the following sentence to the Limitations paragraph: Individual level variability among infants in key covariables, such as nutritional status, may have obscured the magnitude of some findings in this small sample size; we mitigated this by using linear mixed models includes random effects for individuals and fixed effects for key covariables. 

6. You mentioned that “infants with medical illness were excluded from the study, please give examples.

>> We have added a phrase (Participants and Recruitment Procedures paragraph in Methods) to elaborate: “including heart disease, kidney disease, congenital abnormality, genomic syndromes and severe neurological deficits.”

---

## [Decision Letter · Decision Letter 1]

30 Jan 2024

Early full-day leg movement kinematics and swaddling patterns in infants in rural Guatemala: a pilot study

PONE-D-23-20339R1

Dear Dr. Beth A. Smith,

We’re pleased to inform you that your manuscript has been judged scientifically suitable for publication and will be formally accepted for publication once it meets all outstanding technical requirements.

Kind regards,

Tadashi Ito

Academic Editor

PLOS ONE

Additional Editor Comments (optional):

Reviewers' comments:

Reviewer's Responses to Questions

**Comments to the Author**

1. If the authors have adequately addressed your comments raised in a previous round of review and you feel that this manuscript is now acceptable for publication, you may indicate that here to bypass the “Comments to the Author” section, enter your conflict of interest statement in the “Confidential to Editor” section, and submit your "Accept" recommendation.

Reviewer #1: All comments have been addressed

Reviewer #2: All comments have been addressed

2. Is the manuscript technically sound, and do the data support the conclusions?

Reviewer #1: Yes

Reviewer #2: Yes

3. Has the statistical analysis been performed appropriately and rigorously? 

Reviewer #1: Yes

Reviewer #2: I Don't Know

4. Have the authors made all data underlying the findings in their manuscript fully available?

Reviewer #1: Yes

Reviewer #2: Yes

5. Is the manuscript presented in an intelligible fashion and written in standard English?

Reviewer #1: Yes

Reviewer #2: Yes

6. Review Comments to the Author

Reviewer #1: All the addresses modifications have been done as required from the authors , so I see it’s technically good

Reviewer #2: I would like to thank the authors for their work.This paper may open the door for new research about using wearable sensors to detect growth and development impairments.

7. PLOS authors have the option to publish the peer review history of their article (what does this mean?). If published, this will include your full peer review and any attached files.

Reviewer #1: No

Reviewer #2: No

---

## [Editor Report · Acceptance letter]

21 Feb 2024

PONE-D-23-20339R1 

PLOS ONE

Dear Dr. Smith, 

I'm pleased to inform you that your manuscript has been deemed suitable for publication in PLOS ONE. Congratulations! Your manuscript is now being handed over to our production team.

Kind regards, 

on behalf of

Dr. Tadashi Ito 

Academic Editor

PLOS ONE